# Advancing Generalized Deepfake Detector with Forgery Perception Guidance

### Ruiyang Xia
Xidian University
Shaanxi, China
ryon@stu.xidian.edu.cn

### Dawei Zhou
Xidian University
Shaanxi, China
dwzhou.xidian@gmail.com

### Decheng Liu
Xidian University
Shaanxi, China
dchliu@xidian.edu.cn

### Lin Yuan
Chongqing University of Posts and Telecommunications
Chongqing, China
yuanlin@cqupt.edu.cn

### Shuodi Wang
Chongqing University of Posts and Telecommunications
Chongqing, China
ledi0321888@gmail.com

### Jie Li
Xidian University
Shaanxi, China
leejie@mail.xidian.edu.cn

### Nannan Wang*
Xidian University
Shaanxi, China
nnwang@xidian.edu.cn

### Xinbo Gao
Chongqing University of Posts and Telecommunications
Chongqing, China
gaoxb@cqupt.edu.cn

## ABSTRACT

One of the serious impacts brought by artificial intelligence is the abuse of deepfake techniques. Despite the proliferation of deepfake detection methods aimed at safeguarding the authenticity of media across the Internet, they mainly consider the improvement of detector architecture or the synthesis of forgery samples. The forgery perceptions, including the feature responses and prediction scores for forgery samples, have not been well considered. As a result, the generalization across multiple deepfake techniques always comes with complicated detector structures and expensive training costs. In this paper, we shift the focus to real-time perception analysis in the training process and generalize deepfake detectors through an efficient method dubbed Forgery Perception Guidance (FPG). In particular, after investigating the deficiencies of forgery perceptions, FPG adopts a sample refinement strategy to pertinently train the detector, thereby elevating the generalization efficiently. Moreover, FPG introduces more sample information as explicit optimizations, which makes the detector further adapt the sample diversities. Experiments demonstrate that FPG improves the generality of deepfake detectors with small training costs, minor detector modifications, and the acquirement of real data only. In particular, our approach not only outperforms the state-of-the-art on both the cross-dataset and cross-manipulation evaluation but also surpasses the baseline that needs more than 3× training time.

*Corresponding author.

## CCS CONCEPTS

• **Security and privacy → Social aspects of security and privacy**.

## KEYWORDS

Deepfake Detection, Forgery Perception, Training Process

**ACM Reference Format:**
Ruiyang Xia, Dawei Zhou, Decheng Liu, Lin Yuan, Shuodi Wang, Jie Li, Nannan Wang, and Xinbo Gao. 2024. Advancing Generalized Deepfake Detector with Forgery Perception Guidance. In *Proceedings of the 32nd ACM International Conference on Multimedia (MM '24), October 28–November 1, 2024, Melbourne, VIC, AustraliaProceedings of the 32nd ACM International Conference on Multimedia (MM'24), October 28-November 1, 2024, Melbourne, Australia.* ACM, New York, NY, USA, 10 pages. https://doi.org/10.1145/3664647.3680713

## 1 INTRODUCTION

The development of artificial intelligence not only elevates performance in traditional visual tasks [5, 51] but also gives birth to massive novel and heuristic vision applications [21, 30, 48]. Deepfake, a novel technique used to generate believable media via deep neural networks [34], has quickly developed and aroused social concerns due to the lifelikeness of the generation [11, 27, 66] and the simplicity of usage [7, 60, 63]. To ensure the safety and credibility of publicly-oriented media, in the field of computer vision and multimedia research, deepfake detection methods have recently been proposed to discern the authenticity of media automatically.

Early investigations focus on several deepfake techniques [1, 62, 69]. Although these detectors achieve promising performance, their vulnerabilities are exposed immediately when facing the media manipulated by various advanced generative models. Maintaining the detection performance across a broad spectrum of deepfake techniques poses the primary challenge [22, 38].

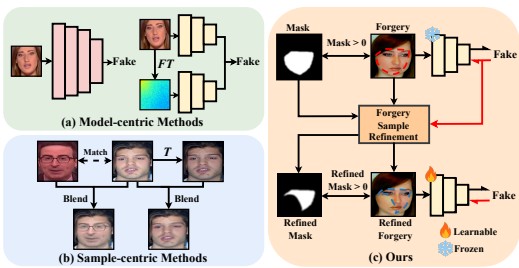

**Figure 1: (a) Model-centric methods increase model complexity or analyze the sample in parallel after Frequency-domain Transformation (*FT*). (b) Sample-centric methods generate forgery samples by blending two different samples after landmark matching or blending a single sample and its variant after Transformation (*T*). (c) Our approach analyzes the perception of the detector to the forgery sample and then refines the sample to improve the forgery perception. The original forgery trace (mask value > 0) and the refined one are respectively denoted with red and blue dashed areas.**

To generalize the deepfake detector, as depicted in Fig. 1(a) and (b), recent works mainly focus on the improvement on model structure and the synthesis of forgery samples. Model-centric methods modify the detector structure based on the public datasets for better perceiving the forgery samples [9, 43, 61, 72], while sample-centric methods deliberately manipulate the common forgery traces such as boundary inconsistency to encourage the detector perceiving these forgery traces [6, 28, 42]. Although there are enhancements in the generalization, the trade-offs between costs and benefits have been overlooked. To be specific, the improved generalization entails integrating complex modules or constructing multiple parallel networks for the model-centric methods. In the case of sample-centric methods, enhancements encompass expensive training costs such as multiple training stages or lengthy training time. Therefore, the efficiency problem motivates us to consider a methodology that elevates the generalization of deepfake detectors with minor detector modifications and small training costs.

To efficiently generalize the detector, one intuitive strategy is to investigate the deficiencies of the forgery perceptions, including the feature responses and prediction scores for forgery samples, and immediately feedback to the detector during training. Based on the insight, an efficient method called Forgery Perception Guidance (FPG) is proposed from the perspective of real-time perception analysis. As depicted in Fig. 1(c), after the gradient computation through backpropagation, the sample is refined to emphasize the forgery traces that were previously unperceived. Subsequently, the detector is trained using the refined sample to strength its perception ability. More specifically, the gradients with respect to both the detector and inputs are respectively converted to the feature response and the adversarial perturbation for refining the shape and magnitude of the forgery traces. As a result, the refined forgery samples not only include forgery traces such as color mismatching but also reflect the deficiencies of the forgery perceptions. This training paradigm pertinently improves the perception ability related to the forgery samples and thus decreases the training costs.

Furthermore, consider the input samples encompass irrespective factors, which are biased to the detectors and affect the perceiving of forgery traces. FPG additionally collects information on image qualities and forgery masks for extra explicit optimizations of the detector. Consequently, the generalization can be improved through FPG with less training costs, minor detector modifications, and the acquirement of real samples only. Experimental results show that FPG is not only superior to state-of-the-art on multiple evaluations but surpasses the baseline that needs more than 3× training time as well. These results demonstrate the high efficiency of our approach. The contributions are summarized as follows:

- We shift the focus to the real-time analysis of the forgery perceptions and elevate the generalization of deepfake detectors through FPG with less training costs, minor detector modifications, and real sample acquirements only.
- FPG refines the forgery samples by modifying the shape and magnitude of the forgery traces based on the feature responses and prediction scores. Moreover, explicit optimizations are further adopted during training by considering the image qualities and forgery masks, respectively.
- Extensive experiments reveal that FPG significantly generalizes deepfake detectors under unknown datasets and deepfake techniques. Moreover, the training time can be saved around 3× compared with the comparable baseline method.

## 2 RELATED WORK

### 2.1 Deepfake Generation

The face generation with malicious intention mainly includes identity replacement and expression reenactment. Identity replacement swaps the face of the target person in a video with the face of the source person. Early techniques focus on one-to-one identity replacement. For instance, FaceSwap uses autoencoders trained on faces with two persons for identity replacement [57]. The following techniques expand the paradigm to many-to-many identity replacement [7, 27, 58]. FaceShifter adopts Adaptive Attentional Denormalization layers (AAD) to transfer localized feature maps between the faces [27]. SimSwap increases the realism by injecting more identity information [7]. In addition, expression reenactment modifies the expression of the source person as arbitrary as the attacker wants. To be specific, the work in [3] applies an improved CycleGAN with two receptive field discriminators to execute one-to-one expression reenactment. Also, some works consider many-to-many expression reenactment [35, 52, 71]. ImaGINator not only considers the fusion between emotion and content but also adopts 3D convolutions to capture the distinct spatio-temporal relationships [52].

### 2.2 Deepfake Detection

**Model-centric Methods.** Since the deepfake techniques mainly manipulate the local regions within the existing real images [34, 49]. To achieve automatic detection, deepfake detectors should carefully observe the image spatial regions for the sake of finding the nuanced artifacts generated through deepfake techniques. Besides, due to the lack of regular textures and semantic information in the forgery traces, conventional networks [10, 17, 44, 45] may not appropriate to deepfake detection. Therefore, some methods aim to improve the detection performance from the detector modifications. Specifically,

a series of novel self-attention mechanisms [36, 64], feature pyramid strategies [4, 14], and graph network blocks [23, 53] are proposed in previous works to elevate the extraction ability of spatial forgery features for the networks. To further explore the samples, the other methods design the parallel networks which can simultaneously analyze the spatial and frequency information [8, 26, 39].

However, most datasets only provide coarse supervision such as binary labels, which makes it difficult for detectors to clearly learn the perceptions of forgery traces. Moreover, the perceptions learned from the previous datasets may be incompatible with the advanced deepfake techniques. Some methods can not achieve great detection performance under the newly generated forgery samples. **Sample-centric methods.** Although multiple deepfake techniques have appeared in recent years, their manipulation pipelines are similar. For example, identity replacement involves the generation of the source face and replacing the target face with blending. Some detection methods thus focus on sample synthesis to remain the common forgery traces like blending artifacts and avoid the traces generated from specific techniques. Besides, more information can be supplied, such as the forgery masks [42], boundaries [28], and bounding boxes [14]. Consequently, the generalization is improved with few [28] or even none [42] detector modifications.

## 3 FORGERY PERCEPTION GUIDANCE

### 3.1 Preliminary

Following the selection of a real sample, the attacker employs a deepfake technique to generate the designated content. This content is defined as the forgery trace, while the operational area of the deepfake technique can be succinctly represented with a forgery mask. Therefore, the essence of deepfake detection is to perceive these forgery traces, and the goal of generalizing a deepfake detector is to cultivate a strong perception ability for arbitrary forgery samples. This entails significant alignments between feature responses and forgery masks, along with notable consistencies between prediction scores and associated labels. To improve the perception ability of the detector, after the analysis of the forgery perception, the discrepancies of the consistencies and alignments should immediately feedback to the detector during training. An intuitive feedback path involves modifying the forgery samples based on these discrepancies, thereby encouraging the detector to learn the forgery traces that have been previously unperceived. However, this modification needs the same deepfake technique on the original forgery samples, which is complex and brings special forgery traces that are adverse to the generalization. To ease the generation of the original forgery samples and ensure these samples reflect the common forgery traces from the deepfake techniques, inspired by [42], the forgery samples are generated as follows:

$$x' = T(x) \odot M + x \odot (1 - M),\qquad(1)$$

where $x' \in \mathbb{R}^{3 \times H \times W}$ indicates the forgery sample, which is generated by blending the real sample $x$ and its variant $T(x)$ after geometric transformations. $H$ and $W$ denotes the height and width of the image. $M \in \mathbb{R}^{3 \times H \times W}$ denotes the forgery mask. The forgery traces of the $x'$ within the $M > 0$. According to Eq. (1), the learned perception of the detector from the forgery samples depends on the geometric transformations and the variations of the forgery

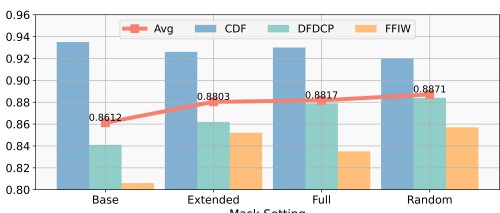

**Figure 2: Cross-dataset evaluation with different mask settings. 'Base' denotes the forgery mask generated based on the facial landmarks related to the jaw, nose ridge, and eyes. 'Extended' is the extension of the eyebrow points up the forehead. 'Full' indicates the whole facial region. 'Random' means randomly selecting one of the mask settings.**

mask. Since $T$ involves amount of discrete parameters, transformations, and non-differentiable during the detector training, we convert to explore the impacts from different forgery mask settings. From Fig. 2, the optimal average result arises from the random mask setting, underscoring the diversities of forgery mask during training. This observation inspires the refinement of the forgery samples from the modifications of the forgery mask. Lastly, the cross-entropy loss is used to measures the difference between the prediction and the associate label of the input sample for guiding the training direction, which is expressed as follows:

$$\mathcal{L}_{CE}(s_i) = - \left[ y_{s_i} \cdot \log\left(p_{s_i}\right) + \left(1 - y_{s_i}\right) \cdot \log\left(1 - p_{s_i}\right) \right],\quad(2)$$

where $y_{s_i}$ and $p_{s_i}$ denote the binary label and the probability being fake for $i$-th sample $s_i$ ($s_i$ may be real or fake), respectively.

### 3.2 Forgery Sample Refinement

As shown in Fig. 3, this procedure entails refining the shape and magnitude of the forgery traces by timely analyzing the forgery perceptions of the deepfake detector. The goal is to adopt pertinent training, thereby elevating the generalization with high efficiency. **Forgery Shape Refinement.** Since the feature response is closely related to the interested objects [41, 68], the perceived forgery traces can be discerned through a comparative analysis of the distinctions between the responses and forgery masks. Given $i$-th fake sample $x'_i$, the weight $\alpha_i^c$ to the feature $F_{x'_i} \in \mathbb{R}^{C \times H' \times W'}$ in the last backbone layer of the detector at the $c$-th channel is computed as:

$$\alpha_i^c = \frac{1}{Z} \sum_u^{H'} \sum_v^{W'} \frac{\partial \mathcal{L}_{CE}(x'_i)}{\partial F_{x'_i(u,v)}^c},\qquad(3)$$

where $Z$ is the spatial size of the feature map. $H'$, $W'$, and $C$ denote the height, width, and channel of the feature map. The feature response $R \in \mathbb{R}^{H' \times W'}$ of the $i$-th sample is computed as follow:

$$R_i = \text{ReLU}\left( \sum_c^C \left( \alpha_i^c \cdot F_{x_i}^c \right) \right).\qquad(4)$$

As depicted in Fig. 3, the shape refinement involves retaining the unperceived forgery traces while reverting the perceived regions to their real counterparts. After generating the feature response through Eq. (3) and (4), $R_i$ is resized to the sample size and compared with the $i$-th forgery mask $M_i$. Given the potential differences in

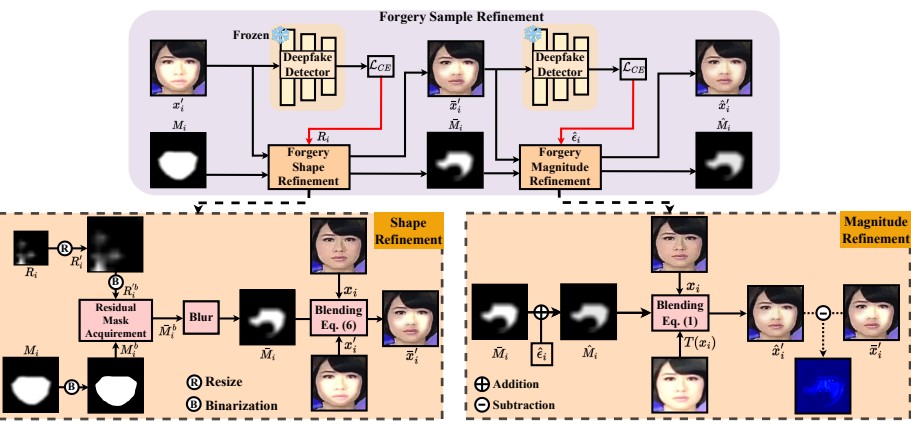

**Figure 3: Overview of the Forgery Sample Refinement. Given the forgery sample $x'_i$, The detector is frozen and the feature response $R_i$ is computed to refine the shape of the forgery traces. After the shape refinement, the perturbation $\hat{\epsilon}_i$ of the mask $\bar{M}_i$ is computed to further conduct the magnitude refinement. To the forgery shape refinement, the residual mask $\bar{M}_i$ related to the unperceived forgery trace is acquired, and the perceived forgery traces are reverted to their real counterparts. To the forgery magnitude refinement, $\bar{M}_i$ is added with the $\hat{\epsilon}_i$ to adjust the magnitude. The $\hat{x}'_i$ is computed by blending the corresponding real image and its variant $T(x_i)$ after geometric transformations. The discrepancy between the shape-refined sample $\bar{x}'_i$ and the $\hat{x}'_i$ with both the shape and magnitude refinement is highlighted in the saliency map.**

orders of magnitude between feature responses and forgery masks, if the magnitude at each location for both the resized responses and masks exceeds 0, it will be binarized to 1. The binary mask with respect to the unperceived forgery trace is thus computed as:

$$\bar{M}^b_i = M^b_i - M^b_i \odot R'^b_i, \tag{5}$$

where for the $i$-th forgery sample, with the binarization, $\bar{M}^b$, $M^b$, and $R'^b \in \{0,1\}^{3 \times H \times W}$ denote the residual mask of the unperceived trace, original forgery mask, and the resized feature response.

However, using the $\bar{M}^b$ directly leads to the noticeable boundary inconsistency in the refined forgery samples. To smooth the boundary, we adopt the GaussianBlur transformation to get the forgery mask $\bar{M}$. Consequently, the sample $\bar{x}'_i$ with forgery shape refinement is generated as follow:

$$\bar{x}'_i = \left(1 - \bar{M}_i\right) \odot x_i + \bar{M}_i \odot x'_i. \tag{6}$$

A straightforward method entails refining each forgery sample throughout the detector training process. Nevertheless, due to the diversities of forgery traces, it is important to notice the potential variability of the forgery perceptions. To the original hard forgery samples, the detector may still not predict them accurately in the early stage. Fewer forgery traces in these samples after refinement will further increase the training difficulties. To the original easy forgery samples, the resized feature responses are close to the forgery masks, the corresponding refined forgery samples will contain fewer forgery traces and are similar to the real counterparts. In the extreme case, $\bar{x}'_i$ will approximate $x_i$ if $R'^b_i = M^b_i$, which is also adverse to the training. Therefore, to make the strategy adaptively refine the forgery samples, the refinement probability is dependent on the sample prediction score, which means the refinement to the hard original forgery samples can be lower than the easy ones. Besides, a forgery sample will be refined if the overlapped size

---

**Algorithm 1** Forgery Shape Refinement

**Input:** $B$: the batch size; $x'_1, \ldots, x'_B$: batch of original forgery samples; $p_{x'_1}, \ldots, p_{x'_B}$: batch of original forgery sample predictions; $M_1, \ldots, M_B$: batch of original forgery masks; $x_1, \ldots, x_B$: batch of corresponding real samples;

**Output:** $\bar{x}'_1, \ldots, \bar{x}'_B$: batch of shape-refined forgery samples; $\bar{M}_1, \ldots, \bar{M}_B$: batch of shape-refined forgery masks;

  **for** $i = 1$ **to** $B$ **do**
    Get the refinement label $o_i \sim \text{Bernoulli}(p_{x'_i})$;
    $M^b_i \leftarrow \text{Binarization}(M_i)$;
    Compute the response $R_i$ through Eq. (3) and Eq. (4);
    $R'_i \leftarrow \text{Resize}(R_i)$;
    $R'^b_i \leftarrow \text{Binarization}(R'_i)$;
    **if** $o_i = 1$ and $\text{Overlap}(R'^b_i, M^b_i) < t$ **then**
      $\bar{M}^b_i \leftarrow M^b_i - M^b_i \odot R'^b_i$;
      $\bar{M}_i \leftarrow \text{GaussianBlur}(\bar{M}^b_i)$;
      $\bar{x}'_i \leftarrow \left(1 - \bar{M}_i\right) \odot x_i + \bar{M}_i \odot x'_i$;
    **else**
      $\bar{x}'_i \leftarrow x'_i, \bar{M}_i \leftarrow M_i$;
    **end if**
  **end for**

---

between the $R'^b_i$ and $M^b_i$ lower than a preset threshold $t$. The whole implementation is summarized in Algorithm 1.

**Forgery Magnitude Refinement.** Inspired by the adversarial attack researches which introduce imperceptible adversarial perturbations thereby causing erroneous predictions [15, 24, 32]. Since these perturbations are closely related to the discrepancies between the prediction scores and the corresponding labels, adding these perturbations to the samples can enlarge the discrepancies and encourage the detector to further reduce the discrepancies during

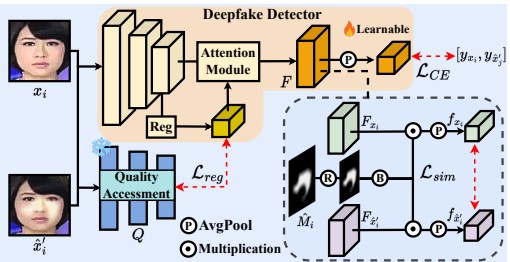

**Figure 4: Overview of the Extra Explicit Optimizations. The detector is unfrozen and updated by inputting the real sample $x_i$ and the forgery sample $\hat{x}'_i$. To further adapt the sample diversities, the detector is optimized by additionally introducing the image quality information as the attention to the detector. Besides, the difference within the local region between the $x_i$ and $\hat{x}'_i$ is also strengthened through $\mathcal{L}_{sim}$. 'Reg' denotes the extra regression branch. The circle 'R' and 'B' have already been described in Fig. 3.**

training, which is suggested to improve the forgery perceptions pertinently. Following the insight, the work in [55] introduces adversarial perturbations as the parameters of GaussianBlur transformation and elevates the generalization of deepfake detection with a large margin. As we focus on refining the forgery mask, for the $i$-th shape-refined sample $\bar{x}'_i$, we further compute the gradient with respect to the forgery mask $\frac{\partial \mathcal{L}_{CE}(\bar{x}'_i)}{\partial \bar{M}_i}$. After that, $\bar{M}_i$ is added with the perturbations computed from the gradients to refine the magnitude of the forgery mask. However, the inherent irregularities from gradient computations will lead to the refined forgery samples being unnatural and inconsistent with the real-world forgery samples. Considering both the adversarial property and the similarity with the real-world forgery samples, for the $i$-th forgery mask $\bar{M}_i$, we collect the gradients with sign function from each pixel and compute a unifying perturbation as follows:

$$\hat{\epsilon}_i = \begin{cases} \epsilon, & \text{if } \sum_u^{H'} \sum_v^{W'} \text{sign}\left(\frac{\partial \mathcal{L}_{CE}(\bar{x}'_i)}{\partial \bar{M}_{i(u,v)}}\right) > 0 \\ -\epsilon, & \text{otherwise} \end{cases}, \qquad (7)$$

where $\epsilon$ is the magnitude of the perturbation. Since we focus on the regions within the forgery traces, the perturbations should not affect the real regions. The forgery mask added with the unifying perturbation at the $(u, v)$-th location is expressed as follows:

$$\hat{M}_{i(u,v)} = \text{Clip}\left(\bar{M}_{i(u,v)} + \hat{\epsilon}_i \bar{M}_{i(u,v)}, 0, 1\right), \qquad (8)$$

where Clip is the truncation function that constrains the magnitude of the $\hat{M}_i$ within [0, 1]. Then the forgery sample $\hat{x}'_i$ with both the shape and magnitude refinement is computed by following Eq. (1).

Moreover, considering the training difficulties if the process includes the adversarial forgery samples only, the adversarial perturbation is added to the forgery mask with a 0.5 probability.

### 3.3 Extra Explicit Optimization

Considering the sample diversities, certain attention modules are employed to make the detector adaptively focus on the forgery

traces. However, these modules are updated based on the binary labels only, which are implicit and lack additional explicit supervision. This implicitness hampers the improvement in forgery perception. Therefore, two extra supervisions are introduced as explicit optimizations for the detector, i.e., image quality and forgery mask.

Given a forgery sample, various image qualities can yield disparities within the forgery traces. It becomes essential to adjust the detector to accommodate such quality variations. In Fig. 4, based on the advanced quality assessment network related to the facial images [37], an extra regression network branch is built into the detector to output the regression results which are supervised by the output of the quality assessment network as follows:

$$\mathcal{L}_{reg}(s_i) = \|\text{Reg}(s_i) - Q(s_i)\|_2^2, \qquad (9)$$

where $Q$ is the pretrained quality assessment network. $\|\cdot\|_2$ denotes Euclidean norm. Reg is the extra regression branch that consists of several convolutional layers. Subsequently, a preset lightweight attention module called Squeeze-and-Excitation [20] is supervised by the image quality information directly, which enables the detector to search for optimal detection patterns for the samples exhibiting diverse image qualities. The details of the regression branch and the attention module are described in the supplementary material.

Furthermore, considering the differences between the $x_i$ and the corresponding fake sample $\hat{x}'_i$ located in $\hat{M}_i > 0$, contrasting the differences of the paired samples is beneficial to the detector perceiving the forgery traces more directly. The mask $\hat{M}_i$ is hence introduced to extract the local features within the forgery region.

Specifically, the $\hat{M}_i$ is resized to the feature size and binarized to get $\hat{M}_i^{'b} \in \{0, 1\}^{C \times H' \times W'}$. After getting the feature of $x_i$ and $x'_i$ from the detector as $F_{x_i}$ and $F_{\hat{x}'_i}$, the feature within the local forgery region are extracted by multiplying $\hat{M}_i^{'b}$. Then the extracted local features are averaged from the spatial dimension to get $f_{x_i}$ and $f_{\hat{x}'_i} \in \mathbb{R}^{C \times 1}$, respectively. An extra loss function $\mathcal{L}_{sim}$ is adopted to the detector for enlarging the difference between $f_{x_i}$ and $f_{\hat{x}'_i}$ through cosine similarity:

$$\mathcal{L}_{sim}(x_i, \hat{x}'_i) = \frac{1}{2}\left(1 + \frac{f_{x_i} \cdot f_{\hat{x}'_i}}{\|f_{x_i}\|_2 \cdot \|f_{\hat{x}'_i}\|_2}\right). \qquad (10)$$

Based on these extra explicit optimizations, the whole loss function during the detector training can hence be expressed as follow:

$$\mathcal{L} = \frac{1}{B} \sum_{i=1}^{B} \left(\mathcal{L}_{CE}(x_i) + \mathcal{L}_{CE}(\hat{x}'_i) + \right. \qquad (11)$$
$$\left. \mathcal{L}_{reg}(x_i) + \mathcal{L}_{reg}(\hat{x}'_i) + \alpha \mathcal{L}_{sim}(x_i, \hat{x}'_i)\right),$$

where $B$ indicates the batch size, and $\alpha$ denotes the scale factor.

## 4 EXPERIMENTS

### 4.1 Settings

**Dataset.** To evaluate the generalization of deepfake detectors, our experiments are conducted on multiple large-scale and widely used datasets: FaceForensics++ (FF++) [40], DeepfakeDetection (DFD) [18], CelebDF (CDF) [29], Deepfake Detection Challenge (DFDC) [19], preview version of DFDC (DFDCP) [13], and Face Forensics in the Wild (FFIW) [70]. FF++ is a large-scale dataset comprising 720

**Table 1: Cross-dataset evaluation on CDF, DFD, DFDC, DFDCP, and FFIW. The results from previous methods are collected from their paper statements. FPG is trained on real samples only from FF++(c23). † indicates the results are performed by [42]. The best and second-best detection results are represented in bold and underline, respectively.**

| Method | Venue | Input Type | Training Set | | Test Set AUC(%) | | | | | |
|---|---|---|---|---|---|---|---|---|---|---|
| | | | Real | Fake | CDF | DFD | DFDC | DFDCP | FFIW | All |
| Face X-ray + BI [28] | CVPR'20 | Frame | ✓ | - | - | 93.47 | - | 71.15 | - | - |
| PCL + I2G [65] | ICCV'21 | Frame | ✓ | - | 90.03 | 99.07 | 67.52 | 74.37 | - | - |
| UIA-ViT [72] | ECCV'22 | Frame | ✓ | ✓ | 82.41 | 94.68 | - | 75.80 | - | - |
| SBI [42] | CVPR'22 | Frame | ✓ | - | 93.18 | 97.56 | 72.42 | 86.15 | 84.83 | 86.83 |
| UCF [61] | ICCV'23 | Frame | ✓ | ✓ | 82.40 | 94.50 | - | 80.50 | - | - |
| L&V [43] | MM'23 | Frame | ✓ | ✓ | 86.00 | 95.50 | - | 83.50 | - | - |
| SeeABLE[25] | ICCV'23 | Frame | ✓ | - | 87.30 | - | **75.90** | 86.30 | - | - |
| AUNet [2] | CVPR'23 | Frame | ✓ | - | 92.77 | **99.22** | 73.82 | 86.16 | 81.45 | 86.68 |
| Two-branch [33] | ECCV'20 | Video | ✓ | - | - | 93.47 | - | 71.15 | - | - |
| DAM [70] | CVPR'21 | Video | ✓ | ✓ | 78.26 | 89.24 | - | 76.53 | - | - |
| LipForensics [16] | CVPR'21 | Video | ✓ | ✓ | 79.40 | 91.90 | - | 79.70 | - | - |
| FTCN [67] | ICCV'21 | Video | ✓ | ✓ | 86.90 | 94.40† | 71.00† | 74.00 | 74.47† | 80.15 |
| TALL+EffNetB4 [59] | ICCV'23 | Video | ✓ | ✓ | 83.37 | - | - | 67.15 | - | - |
| AltFreezing [54] | CVPR'23 | Video | ✓ | ✓ | 89.50 | 98.50 | - | - | - | - |
| FPG | - | Frame | ✓ | - | **94.49** | 96.41 | 74.75 | **87.24** | **87.93** | **88.16** |

training videos, 140 validation videos, and 140 test videos. The fake videos are produced by two identity replacement techniques, Deep-Fakes (DF) [56] and FaceSwap (FS) [57], as well as two expression reenactment techniques, Face2Face (F2F) [47] and NeuralTexture (NT) [46]. We use the fake videos from the FF++ dataset for evaluating cross-manipulation performance. Following previous works [4, 39, 53], the c23 version of FF++ is adopted for training. More-over, since the CDF-v2 is more challenging than the v1 version, the results in CDF come from the v2 version in default.

**Implementation Details.** Each pristine video is sampled 8 frames as the training sample. By default, we employ EfficientNetB4 [45] as the deepfake detector, which is initialized through pre-training on the ImageNet [12]. The batch and image size are the same as the baseline settings [42]. The threshold $t$ and $\epsilon$ in forgery sample refinement is set to 0.2 and 0.01, respectively. Empirically, the scale factor $\alpha$ is set as 0.075. The maximum of training epochs is 50. The learning rate equals 1e-4. The optimizer is AdamW [31] with 0.9 and 0.98 betas, alongside 0.2 weight decay. The experiments are conducted on four RTX 3090 GPUs and the Pytorch framework.

**Evaluation.** Following previous works in [2, 16, 42], we use video-level predictions to evaluate detectors. Each video are predicted through sampling 32 frames and averaging their predictions for comprehensive evaluation. We use the Average Precision (AP) and Area Under the Receiver Operating Characteristic Curve (AUC) as the metrics to evaluate the detection performance.

## 4.2 Generalization Performance Evaluation

**Cross-Dataset Evaluation.** The generalization constitutes the primary determinant for a deepfake detector. Since the public forgery datasets involve multiple deepfake techniques, complex real-world scenes, and various personal characteristics, the detection results

across multiple datasets hold significant reference value for assessing the generalization performance. Therefore, we first evaluate the AUC results under five prevalent and challenging datasets that remain unknown to the deepfake detector. Table 1 presents our cross-dataset evaluation results. Compared with previous promising methods, the detector with FPG is trained on real samples only and outperforms the second-best method on CDF, DFDCP, and FFIW by 1.31%, 1.08%, and 3.10% points, respectively. To the DFD, we can also achieve a comparable result. As a result, leveraging the refinement strategy and the explicit optimizations, our approach attains 88.16 % AUC on average for five datasets and surpasses the baseline method [42] by 1.33% (88.16 % vs. 86.83%). These outcomes demonstrate the effectiveness of our approach.

**Cross-Manipulation Evaluation.** Given the difference in deep-fake techniques significantly affects the generalization, the next experiments are hence conducted to evaluate the detection results under different deepfake techniques. We follow the previous protocols [2, 25, 28, 42, 54, 65] by using the raw version of FF++ for evaluation. To be specific, our approach is evaluated through four deepfake techniques: DF, F2F, FS, and NT. Similar to the above evaluation, we compare the performance with both the video-level and frame-level methods. The evaluation results are listed in Table 2. As seen, the results evaluated on DF and FS nearly equal the best performance (with only 0.02% and 0.25% AUC discrepancy). On the other hand, the detector attains the best results on F2F and NT and outperforms state-of-the-art methods on average. The promising detection results exhibited by the detector underscore the enhancements in generalization achieved by FPG.

## 4.3 Ablation Study

**Training Efficiency Analysis.** As stated, FPG is proposed to not only elevate the generalization of deepfake detectors but also the

**Table 2: Cross-manipulation evaluations on DF, F2F, FS, and NT. ‡ indicates the results are reimplemented on FF++(c23) through the official code. The best and second-best detection results are represented in bold and underline, respectively.**

| Method | Venue | Test Set AUC | | | | |
|---|---|---|---|---|---|---|
| | | DF | F2F | FS | NT | Avg |
| Face X-ray + BI | CVPR'20 | 99.17 | 98.57 | 98.21 | 98.13 | 98.52 |
| PCL + I2G | ICCV'21 | **100** | 98.97 | 99.86 | 97.63 | 99.11 |
| SBI‡ | CVPR'22 | 99.98 | 99.80 | 99.42 | 98.07 | 99.32 |
| SeeABLE | ICCV'23 | 99.20 | 98.80 | 99.10 | 96.90 | 98.50 |
| AUNet | CVPR'23 | 99.98 | 99.60 | **99.89** | 98.38 | 99.46 |
| AltFreezing | CVPR'23 | 99.80 | 98.20 | 99.70 | 96.20 | 98.48 |
| FPG | - | 99.98 | **99.83** | 99.64 | **98.46** | **99.48** |

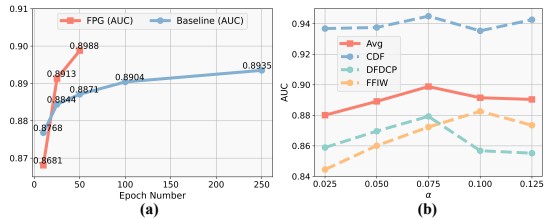

**Figure 5: Detection performance on multiple unknown datasets. (a) The comparison between our approach and the baseline is based on averaged detection results obtained from the CDF, DFDCP, and FFIW datasets. (b) The variation of the detection performance with respect to the scale factor $\alpha$.**

**Table 3: Ablation study of different component settings. 'B', 'R', and 'O' indicate baseline, forgery sample refinement, and extra explicit optimizations. AP and AUC are averaged from the results on CDF, DFDCP, and FFIW. The bracket of 'H' and 'M' denotes hour and million bytes, respectively.**

| Method | | | Epoch | AP (%) | AUC (%) | Time (H) | Param (M) |
|---|---|---|---|---|---|---|---|
| B | R | O | | | | | |
| ✓ | - | - | 50 | 90.33 | 88.71 | **1.6** | **16.7** |
| ✓ | - | - | 250 | 91.86 | 89.35 | 8.3 | 16.7 |
| ✓ | ✓ | - | 50 | 91.72 | 89.27 | 2.5 | 16.7 |
| ✓ | ✓ | ✓ | 50 | **92.48** | **89.88** | 2.6 | 18.2 |

**Table 4: Ablation study of different detectors. The AUC results are tested from CDF, DFDCP, and FFIW. ‡ indicates the results are implemented on FF++(c23) through the official code since [42] does not support VGG-19 results.**

| Backbone | Test Set AUC (%) | | | |
|---|---|---|---|---|
| | CDF | DFDCP | FFIW | Avg |
| Xception + Baseline [42] | 90.27 | 78.85 | 76.72 | 81.95 |
| Xception + FPG | **90.68** | **83.57** | **78.98** | **84.41** |
| ResNet-50 + Baseline [42] | 90.66 | 82.88 | 79.30 | 84.28 |
| ResNet-50 + FPG | **90.72** | **82.93** | **81.77** | **85.14** |
| VGG-19 + Baseline‡ | 81.17 | 75.07 | 82.54 | 79.59 |
| VGG-19 + FPG | **82.31** | **75.94** | **85.07** | **81.11** |

training efficiency. To analyze the efficiency in detail, the detector with FPG is trained with three different epochs, i.e., 10, 25, 50. As comparison, the baseline is also trained with the corresponding epochs and the same training settings for fairness. Fig. 5(a) lists the average AUC results on CDF, DFDCP and FFIW. As seen, since the diversities of forgery traces are increased through FPG, too short training epochs make the detector under-fitting and insufficient for generalization. However, the advantage of utilizing FPG appears apparent when continues to increase the epoch. When the epoch number equals 50, all metrics from our approach surpass the baseline with significant margins. To further explore the number of epochs required for the baseline to approximate the performance achieved by FPG, we extend the maximum of training epochs to 100 and 250, respectively. Apparently, the improvement rate in detection results is sluggish. We conjecture the reason is the lack of a compact correlation between forgery sample generation and the cultivation of forgery perception. Consequently, using the original forgery samples to cultivate a comprehensive forgery perception proves to be inefficient. Even when the epoch number equals 250, the average detection results with the baseline method remain inferior to FPG trained with only 50 epochs.

Table 3 presents more details related to the detection performance and the costs. As we bridge the forgery sample generation and the cultivation of forgery perception through the FPG. From the third row of Table 3, after adopting the refinement strategy

only, the generalization performance can be effectively improved with the same detector structure. Besides, since the robustness of the detector to the irrespective factors within the input samples is also important to the detection results, we implement extra explicit optimizations to adapt sample diversities by considering image qualities and local forgery traces. Consequently, the generalization is further improved with only 1.5M parameter increasing (18.2 vs. 16.7). Remarkably, our approach substantially reduces training cost, saving around 3× training time compared to the baseline method (8.3 vs. 2.6), underscoring the high efficiency of our approach.

**Method Applicability Analysis.** To evaluate the effectiveness of FPG on the other deepfake detectors, i.e., Xception [10], ResNet50 [17], and VGG [44]. These detectors are trained with the baseline method and FPG, respectively. As shown in Table 4, with the help of the refinement strategy and explicit optimizations, FPG outperforms the baseline method on multiple detectors. For example, FPG can elevate the detection results of Xception to 84.41%, which surpasses the baseline method by over 2.46%. Moreover, these results produced by FPG can effectively demonstrate its applicability, which is important in scenarios where the computing source, technique compatibility, or model scalability are constrained. Moreover, the experiments related to the applicability of different image distortions are listed in the supplementary material.

**Hyperparameter Setting Analysis.** We first analyze the impact brought by the scale factor $\alpha$. Fig. 5(b) lists the detection results on CDF, DFDCP and FFIW. $\alpha$ is used to measure the strength of the direct contrast between a real sample and the fake counterpart

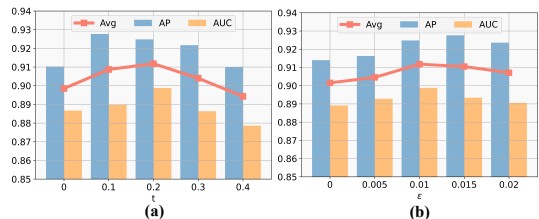

**Figure 6: The variation of Detection results. (a) The impact with different $t$. (b) The impact with different $\epsilon$.**

in the local forgery traces. Increasing the $\alpha$ from 0.025 to 0.075 leads to an improved detection performance. This is because the direct contrast of the local features elevates the sensitivity of the detector to the forgery traces. The decline appears when $\alpha$ beyond 0.075. This may be attributed to the respective field of the feature map. $\mathcal{L}_{sim}$ uses the local features that are selected from the resized forgery mask. Since these deep layer features possess high receptive fields, indicating that the selected features related to the image area outside local forgery regions and increase the $\alpha$ thus enlarging the biases. To get correlated features, a more precise feature selection strat egy will be considered in our future work.

In the forgery sample refinement, $t$ is the overlapped threshold between the forgery masks and the corresponding resized feature responses. $\epsilon$ involves the modifications to the magnitude within the forgery trace. To investigate the impact of the hyperparameter, as depicted in Fig. 6(a), the upward trend appears when $t$ is increased from 0 to 0.2, which means the forgery samples generated through the shape refinement can reflect the detector omissions to the forgery traces and then conduct more efficient training. Further increasing leads to fewer forgery traces within the samples, exacerbating the detection difficulties and hard to converge during training. Fig. 6(b) illustrates the detection result related to $\epsilon$. Similarly, increasing the perturbation improves the detection result. Nevertheless, since the magnitude of the mask is adjusted through a one-step attack strategy [15], perturbations are generated by linear approximation of the gradients. This approximation introduces biases and decreases the detection result when $\epsilon$ is further increased.

### 4.4 Qualitative Study

**Forgery Perception Analysis.** To investigate the perception of detector to the forgery samples more intuitively, the saliency map of forgery samples with different deepfake techniques is shown in Fig. 7. It can be apparently observed that, with the help of FPG, there are higher salient values than the baseline method within the forgery traces, which means the detector can perceive the forgery traces more completely. As a result, the deepfake detector can better adapt to the forgery samples with various deepfake techniques so that the generalization can be improved remarkably. More qualitative results are shown in the supplementary material.

**Sample Distribution Analysis.** To make the detector adapt to the sample diversities, extra explicit optimizations are introduced to further consider the facial image qualities and local forgery traces during the detector training. Fig. 8 illustrates the impact brought by

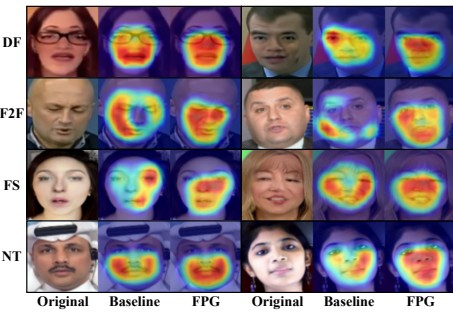

**Figure 7: The saliency visualization for samples with different deepfake techniques. FPG is compared to the [42].**

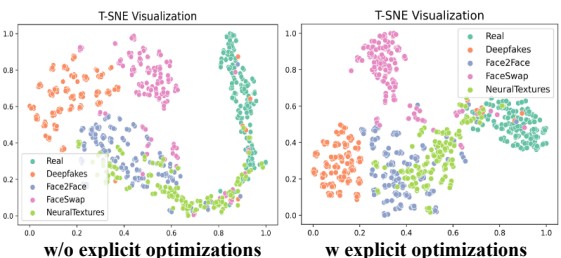

**Figure 8: The T-SNE [50] visualization of our approach without and with the extra optimizations. The distributions of the samples with the same category exhibit more compact by optimizing the detector with more explicit information.**

the optimizations to the sample distributions when facing unknown deepfake techniques. In the absence of explicit optimizations, the distributions of samples with the same class present dramatic variations such as sparse scattered or long trails, suggesting that the detector is sensitive to the irrespective factors. With explicit optimizations, the detector can better adapt the various samples and focus on the features related to the forgery traces. Due to the similar characteristics of samples with the same deepfake techniques, these distributions thus becomes clustered.

## 5 CONCLUSION

In this paper, an efficient method dubbed Forgery Perception Guidance (FPG) is proposed to elevate the generalization of deepfake detectors from the perspective of real-time perception analysis, which has inadequate attention in previous studies. Specifically, FPG encompasses the forgery sample refinement and the extra explicit optimizations. The refinement strategy pertinently facilitates perceiving the forgery traces by modifying the samples based on the feature responses and prediction scores. The extra explicit optimizations promote adapting sample diversities by considering facial image qualities and local forgery masks, thereby further improving the forgery perceptions. Experiments demonstrate that FPG is superior to state-of-the-art methods on the cross-dataset and cross-manipulation protocols. Notably, the enhanced generalization is achieved with small training costs, minor detector modifications, and the acquirement of real samples only.

# 6 ACKNOWLEDGMENTS

This work is supported in part by the National Natural Science Foundation of China under Grant U22A2096, Grant 62441601, Grant 62306227, Grant 62201107, and Grant 62221005; in part by the Fundamental Research Funds for Central Universities under Grant QTZX23042 and Grant ZYTS24142; in part by the Natural Science Foundation of Chongqing under Grant CSTB2022NSCQ-MSX1265; and in part by the Science and Technology Research Program of Chongqing Municipal Education Commission under Grant KJQN202 300606.

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
