# OpenReview forum: "Advancing Generalized Deepfake Detector with Forgery Perception Guidance"
_acmmm.org/ACMMM/2024/Conference — MM2024 Poster_

### Official Review · Reviewer_CVgq · 2024-05-24

**Rating:** 4
**Confidence:** 4

**Summary:**

This paper presents a Forgery Perception Guidance (FPG) based method for face forgery detection. It builds on SBI[1], and proposed to refine the shape and magnitude of the manipulation region to improve the quality of generated images.

**Strengths:**

(1) The motivation is clear, and I think the proposed method is feasible.

(2) The results shown in Tables 1 and 2 are promising.

**Limitations:**

(1) Face X-ray and SBI are conducted on the FF++ raw data, so Table 1 is somewhat not right. As I know, the issue with these blending-based methods is that they do not perform well on compressed data. I have some doubts that the proposed method can achieve this result on the FF++ C23. Though the authors have released their code, I think it is better to provide the evaluation code and the trained model to verify its effectiveness.

(2) Another problem in blending-based methods is their poor robustness performance, can the author provide the robustness performance?

[1] Detecting deepfakes with self blended images, CVPR 2022.

**Suitability:**

2

---

### Official Review · Reviewer_d42N · 2024-05-25

**Rating:** 4
**Confidence:** 4

**Summary:**

In this paper, the authors propose an efficient method, i.e., Forgery Perception Guidance (FPG), to generalized deepfake detectors. Specifically, FPG adopts a sample refinement strategy to improve diversity of synthesized samples via the feedback of the optimization objective. Moreover, FPG introduces more sample information as explicit optimizations, which makes the detector further adapt the sample diversities. Extensive experiments demonstrate the effectiveness of the proposed method.

**Strengths:**

1. The proposed FPG could be exploited by existing method of synthesizing forgery artifacts, e.g., SBI, Face X-ray, and PCL, and further improve their performance.
2. FPG leverages differences of gradients in difference regions to further regenerate more targeted forgery samples, which guides the detector to mine more discriminative artifacts. Furthermore, the authors introduce the frozen quality assessment network with designed loss functions to guarantee the quality of synthesized samples.
3. Extensive experiments illustrate that SBI-based FPG method outperforms most existing methods in terms of cross-dataset and cross-manipulation generalization evaluations.

**Limitations:**

1. RFM (Representative Forgery Mining for Fake Face Detection, CVPR 2021) also exploits the gradient feedback to guide the detector capture more comprehensive artifacts. It is encouraged that the authors compare this method and describe detailed differences. Moreover, in my opinion, FPG could be applied to any synthesis-based methods, e.g., PCL, Face X-ray, etc. I encourage the authors to include more basic methods and demonstrate the effectiveness of FPG, comprehensively.
2. As described in method part, forgery sample refinement exploits the gradient back-propagation of the cross-entropy loss function to regenerate diverse fake samples. However, in early training stage, the detector is unable to provide accurate forgery regions via gradient back-propagation, how the authors tackle this issue? Warm-up the detector with few epochs, or ? I expect the authors provide implementation details.
3. For a good detector, we also expect it to perform well under several real-world perturbations, e.g., Gaussian Noise, compression, color mismatch, etc. Therefore, I encourage the authors to conduct extensive experiments on robustness performance.

**Suitability:**

2

---

### Official Review · Reviewer_dTAv · 2024-05-25

**Rating:** 4
**Confidence:** 4

**Summary:**

The paper deals with deepfake detection (facial images) by resorting to a forgery perception guided approach.

**Strengths:**

The paper addresses an important open issue such as generalization in deepfake detection.
It introduces the idea to force the attention on specific parts of the face related to the specific forgery by using a masking approach.
It only considers real sample for training the forgery perception part.
Experimental results appear to achieve the sota.

**Limitations:**

The work is not so well presented and it is quite difficult for the reader to follow the whole adopted pipeline.
Also the way experiments are carried out is not often so clear.
Furthermore, it is not evident the motivation behind: which is the effective relation between a fake sample generated according to Equation (1) starting from a transformed real one and the generated fake samples obtained through the diverse DF techniques?
Often, it seems that the process also uses fake sample to train, please be careful when adopting the term "forgery sample" or similar.
It is not clear (Table 1), if all the method have been trained in the same way due to the fact that results are taken from their paper statements (see caption of Table 1), please explain this crucial issue? Furthermore, how is the DF detector (EfficientNetB4) trained on FF++? All the different forgeries together?
Moreover, in Table 2 (cross-forgery), same question as before? Table 2,  is it on raw FF++ images (see raw 670) or c23 as said in the caption?

**Suitability:**

2

---

### Official Review · Reviewer_EEkz · 2024-05-27

**Rating:** 4
**Confidence:** 4

**Summary:**

This paper proposes an efficient method to finetune the shape and magnitude of the forgery mask. The author conducted relatively comprehensive experiments to show its efficacy and effectiveness.

**Strengths:**

1. Overall this paper is well-motivated. I think the forgery response is a good starting point to design an algorithm.

2. The paper is well-written and easy for me to understand. There are no apparent logic gaps or truth errors.

3. The experiments conducted in this work are relatively sufficient.

4. This work shows a comprehensive discussion of related works, especially some generation techniques that are not well known in the detection field.

**Limitations:**

A significant limitation of this work is its heavy reliance on forgery masks. All the modules proposed in this work are heavily dependent on these masks, which is why this work uses SBI as their baseline. However, the improvement compared to the baseline (SBI) is minimal, with only about a 1% improvement on CDF-v2. Therefore, the primary contribution to the performance seems to be from SBI itself.

When considering the "efficacy module" proposed in this work, it relies on masks as their "golden annotations," making some supervised learning methods unfeasible and only allowing for SBI-like self-supervised methods. If we need to train on forgery data without any mask label, it would be challenging to apply your method in practice. In reality, we often do not know the forgery mask and its corresponding forgery region in a given deepfake video/image. Moreover, as mentioned in the related work section on generation techniques, most current state-of-the-art generation methods are many-to-many and do not involve a blending process but directly generate a whole image. So, in these cases, does your method still have potential applicability? I understand that many existing detection methods rely on blending data for training instead of deepfake data. However, the concern I've mentioned is crucial and increasingly significant for deepfake detectors. Therefore, I need more clarification on this point.

Another observation is that the refined shape of the mask appears visually strange. Intuitively, the forgery mask should resemble the original SBI's mask, with a convex hull or other shapes. However, after your refinement, the shape does not seem to resemble the actual deepfake forgery mask. So, how does your refinement achieve better performance?

My last minor point pertains to the t-SNE visualization. Based solely on the visualization, I cannot determine why your method is superior to the baseline. Could you provide more insights or explanations regarding this figure?

**Suitability:**

3

---

### Meta-Review · Area_Chair_MQQ2 · 2024-06-25

**Recommendation:** Accept (Poster)
**Confidence:** 5

**Metareview:**

All reviewers rated the paper to be fine, the authors are suggested to take all the comments into considerations for the final version.